# Mechanical Evaluation of the Stability of One or Two Miniscrews under Loading on Synthetic Bone

**DOI:** 10.3390/jfb11040080

**Published:** 2020-11-05

**Authors:** Andrea Pradal, Ludovica Nucci, Nicola Derton, Maria Elena De Felice, Gianluca Turco, Vincenzo Grassia, Luca Contardo

**Affiliations:** 1Department of Medical, Surgical and Health Sciences, University of Trieste, 34127 Trieste, Italy; andreapradal.office@gmail.com (A.P.); info@studiodentisticoderton.com (N.D.); gturco@units.it (G.T.); l.contardo@fmc.units.it (L.C.); 2Multidisciplinary Department of Medical-Surgical and Dental Specialties, University of Campania Luigi Vanvitelli, 81100 Naples, Italy; ludortho@gmail.com (L.N.); merielendf@hotmail.it (M.E.D.F.)

**Keywords:** miniscrews, bone anchorage, primary stability, load value, TADs

## Abstract

The aim of the present study was to evaluate the primary stability of a two-miniscrew system inserted into a synthetic bone and to compare the system with the traditional one. Forty-five bi-layered polyurethane blocks were used to simulate maxillary cancellous and cortical bone densities. Samples were randomly assigned to three groups—one-miniscrew system (Group A, *N* = 23), two-miniscrew system (Group B, *N* = 22) and archwire-only (Group C, *N* = 10). A total of 67 new miniscrews were subdivided into Group A (23 singles) and Group B (22 couples). 30 mm of 19″ × 25″ archwires were tied to the miniscrew. The load was applied perpendicularly to the archwire. Maximum Load Value (MLV), Yield Load (YL) and Loosening Load (LL) were recorded for each group. The YL of Group B and C had a mean value respectively of 4.189 ± 0.390 N and 3.652 ± 0.064 N. The MLV of Group A, B and C had a mean value respectively of 1.871 ± 0.318N, of 4.843 ± 0.515 N and 4.150 ± 0.086 N. The LL of Group A and B had a mean value respectively of 1.871 ± 0.318 N and of 2.294 ± 0.333 N. A two- temporary anchorage device (TAD) system is on average stiffer than a one-TAD system under orthodontic loading.

## 1. Introduction

For a long time, orthodontists have struggled to achieve efficient control over the anchorage. Adequate forces control is essential when treating skeletal and dental malocclusions as unwanted movements of the anchor unit may occur.

With the introduction of miniscrews as temporary anchorage devices (TADs), stable intraosseous anchorage could be achieved. These devices are very small and can be placed in areas where other TADs fail, for example between the roots of individual teeth of both maxillary and mandibular bones [1]. Unlike restorative endosseous implants, miniscrews do not require osseointegration. Moreover, compared with other TADs, miniscrews are less expensive, they are small enough for placement at any surface of the alveolar process and techniques for manipulation are relatively simple [2]. These devices rely on primary stability (mechanical retention), which makes them immediately loadable, simple and less invasive to remove. As orthodontic loading is applied immediately after application, the primary stability, which is mostly mechanical retention, is essential in maintaining a high success rate [3]. Since their retention is mostly mechanical, biomechanical factors must be considered indeed they have a higher chance of loosening due to torque or rotational forces under loading.

Many factors are decisive for their primary stability. These include the insertion angle [4], bone mineral density (BMD) of the receiver site, screw design [3,5,6,7], number of screws used as an anchoring unit.

An ideal miniscrew would require minimal insertion torque so that the screw has low fracture risks and low bone strain; by contrast, the removal torque should be high enough so that it does not easily loosen while loaded.

Efforts to increase the removal torque led to the development of the tapered type of miniscrew, which has a greater diameter near the screw head. According to finite element analysis, the conical shape provides better strength, mechanical stability and less stress at the collar level [8].

Although many studies focused on the axial pull out strength, they did not reflect the constant radial loading used for orthodontic treatments in a clinical setting. Lee et al. suggested that the loading level used for orthodontic treatment can develop micromotion between the miniscrew and surrounding bone, which can reduce its primary stability immediately after implantation. Moreover, the failure risk of the mini-implant system likely increases if more displacement of the miniscrews propagates when peri-implant BMD variability increases due to active bone remodeling during early post-implantation healing and prolonged orthodontic treatment loading periods [9].

Augmented mobility of the miniscrew or loss of stability under orthodontic load leads to failure. Success rates are between 70 and 100% [10,11,12], with an overall failure rate of 13.5% [13,14,15,16]. In the past, different solutions such as using a larger-diameter miniscrew, changing the insertion site [17,18,19] or secondary insertion [20,21] have been analyzed to overcome the loss of stability. We hypothesized that adding a second miniscrew could increase the anchorage reducing the risk of movements under the orthodontic load of the proximal miniscrew.

The aim of the present study was to evaluate the primary stability of a two-miniscrew system inserted into a synthetic bone and to compare the system with the traditional one. To the best of our knowledge, this is the first study that investigated a two-miniscrew system to stiffen the miniscrew anchorage. The bone simulation was limited to a buccal maxillary scenario between the distal aspect of the canines and the mesial aspect of the first molars.

The null hypothesis was that the two systems (single- or two-miniscrew) are mechanically equivalent.

## 2. Materials and Methods

Forty-five bi-layered polyurethane blocks (Sawbones, Pacific Research Laboratories, Vashon Island, WA, USA) were used to simulate maxillary cancellous and cortical bone densities. According to Devlin, the mean bone mineral density is 0.31 g/cm^3^ (*SD* = 0.14) for the posterior maxilla [22]. The cortical thickness and density measurements at 8 mm from the alveolar crest are 0.78 mm (*SD* = 0.16) and 1291 HU (*SD* = 138), respectively [23]. According to the Misch and the Lekholm & Zarb classifications and Sawbones specifics, polyurethane foam of 20 pcf density and a 1-mm polyurethane sheet of 30 pcf density were used to mimic cancellous and cortical bone, respectively. Each bi-layered block measured 10 × 20 × 16 mm.

A total of 67 new self-drilling titanium miniscrews with a diameter of 1.6 mm and a length of 7.5 mm (Quattro BH MINI PLUS, Ortoteam s.p.a., Milan, Italy) were subdivided into two groups: 23 singles (Group A) and 22 couples (Group B).

Rectangular 19″ × 25″ TMA (Ormco Corporation, Orange, CA, USA) archwires were cut into segments of 30 mm and tied to the miniscrew with stainless steel ligatures.

Each TAD was perpendicularly inserted into the polyurethane blocks using a WS75-L contra-angle connected to an Elcomed motor (W&H Dentalwerk Bürmoos GmbH, Salzburg, Austria). The settings for the motor were a uniform speed of 50 rpm and a torque limiter of 10 N∙cm. All insertions were recorded via USB (Table 1). All TADs were inserted until the last thread; for Group B, a space of 10 mm was left between the two TADs. For Group C, a custom-made clamp was designed and printed with an Ultimaker 3 Extended (Ultimaker, Utrecht, The Netherlands) using polylactic acid and polyvinyl acid as copolymer. The resolution of the 3D printer was 0.4 mm. Subsequently, the clamp was fixed on a rigid support and a segment of 30 mm of rectangular 19″ × 25″ TMA archwire were tied within with stainless steel ligatures.

### 2.1. Measurements

Mechanical tests were performed on a universal mechanical testing machine (Galdabini Sun 500, Galdabini, Varese, Italy). The probe speed was maintained constant at 1 mm/min (Figure 1).

The load−displacement profile was recorded through the manufacturer’s software. Samples were randomly assigned to three groups: one-miniscrew system (Group A, *N* = 23), two-miniscrew system (Group B, *N* = 22) and archwire-only (Group C, *N* = 10). The samples were clamped and tested individually. The load was applied perpendicularly to the archwire at a distance of 10 mm from the proximal point of TAD for Groups A and B and from the proximal point of the clamp for Group C (Figure 2).

The probe was in contact with only the larger side of the section of the wire. For Group A and Group B, the samples were positioned in such a way that the applied load would loosen the TAD.

First, the yield load (YL) was found. This parameter is defined as the point on a stress-strain or, as in this case, on the load-displacement curve, which indicates the end of the elastic region and the beginning of the plastic region. Second, a maximum load value (MLV) was established as the highest point reached by each group on the load-displacement curve.

Third, the loosening load (LL) was defined as the load in which the first anticlockwise movement could be observed respectively for the TAD itself in Group A and the proximal TAD in Group B.

### 2.2. Statistical Analysis

The statistical significance of the differences among the data was evaluated using SPSS v.24 for Mac. Due to the rejection of the assumptions of normality and homoscedasticity, the data were compared with nonparametric tests: Kruskal-Wallis and Mann-Whitney tests. The statistical significance level was set at *p* = 0.05. Sample size was calculated trough of G*Power 3.1.9.6 for Mac OS imposing 3 different settings (Group A: one-miniscrew system; Group B: two-miniscrew system; Group C: archwire-only), a single tail, an effect size equal to 0.5, an error probability ratio equal to 0.2 and a power level of 0.8.

## 3. Results

Insertion Torque was recorded for each screw and then the mean value was calculated and reported for each group (Table 1). No statistical differences were found among the insertion torque values of the TAD (Figure 3 and Table 1).

MLV, YL and LL were recorded for each group and are reported in Table 2; the statistical analyses are reported in Figure 3, Figure 4, Figure 5 and Figure 6. The YL of Group A could not be calculated since the archwire maintained an elastic profile throughout the test. The YL of Group B ranged between 3.635 N and 5.109 N with a mean value of 4.189 ± 0.390 N. The YL of Group C ranged between 3.584 N and 3.741 N with a mean value of 3.652 ± 0.064 N. (Figure 4 and Table 2).

The MLV of Group A ranged between 0.812 and 2.524 N with a mean value of 1.871 ± 0.318 N. The MLV of Group B ranged between 3.954 N and 5.862 N with a mean value of 4.843 ± 0.515 N. The MLV of Group C ranged between 4.018 N and 4.260 N with a mean value of 4.150 ± 0.086 N. (Figure 5 and Table 2).

The LL of Group A ranged between 0.812 N and 2.524 N with a mean value of 1.871 ± 0.318 N. The LL of Group B ranged between 1.484 N and 3.061 N with a mean value of 2.294 ± 0.333 N. (Figure 6 and Table 2)

In a system with a single miniscrew (Group A), the YL of the archwire could not be reached because the TAD starts to loosen at a lower load compared with the YL of Group C (archwire). The system with two miniscrews (Group B) is mechanically stiffer than Group A; in fact, the highest MLVs of Group B are twice as large as the MLVs of Group A and they are on average higher than the MLVs of Group C. Although the LL values of Group B are lower than the corresponding MLVs, they are statistically higher than the LL values of Group A (*p* < 0.05). The MLVs of Group B are reached after the YL of the attached archwire. The second miniscrew stiffens the system enough to reach the YL of the attached archwire so that the unscrewing torque of the proximal TAD probably depends on the mechanical features of the archwire.

For the three groups considered in this study, the load-displacement graphs of some representative samples are reported in Figure 7.

## 4. Discussion

The present study suggests the placement of a second miniscrew to increase the mechanical retention of the main miniscrew reducing the risk of failure.

The aim is to evaluate the primary stability and load value of a two miniscrew system compared with a traditional one miniscrew, inserted into a synthetic bone that simulates the cortical and cancellous bone of the buccal aspect of the maxilla.

The null hypothesis was that the two systems are mechanically equivalent.

In agreement with the statistical differences between the two groups, the null hypothesis was rejected.

Concerning the one-miniscrew system, there is a typical elastic graph up to the MLV point, followed by a sawtooth profile and each peak is lower than the previous one, which could correspond to a continuous and stepped loosening of the TAD. Because of the resistance offered by the second miniscrew, this sawtooth profile was not found in the two-miniscrew system, although an initial loosening (LL) can be noticed.

Literature review has showed that the researches, relating to the loss of primary stability of mini-screws, no analyzes the unscrewing induced by the force applied to the wire but performed only pull out [5,11], bend [23] or fracture tests [12,15]. So they are not comparable with what has been analyzed and tested in our work because they had effort on different mechanics.

Therefore, when we consider the primary stability of a mini-screw, we must relate it to different factors such: bone density [6,10], cortical thickness [5,21], the inclination of insertion [7,11] and insertion torque [17]. The influence of many of these factors can, hence, as emerges from our study, be minimized with a two-miniscrew system, where load and torsion forces are best issued.

Although the MLV of a two-miniscrew system can be higher than the orthodontic load usually requested on daily clinical practice, the use of a second miniscrew may reduce the risk of failure when detailed conditions are showed that loosening of a single miniscrew is expected.

What emerges from our results is that often the use of a single screw is not useful to develop a reliable and predictable biomechanical system. Therefore the possibility of using two screws should always be considered [19,20,21]. This, in addition to the biomechanical advantages described, involves limits linked to the need to find two nearby sites suitable for positioning which in turn must be aimed at a single system, therefore allowing the simultaneous engagement of an orthodontic wire [24,25,26].

The unscrewing problem can also be reduced, in our opinion, using clockwise and anticlockwise TADs so that the orthodontic mechanics would not unscrew the TADs, although the armamentarium would be doubled [27,28,29].

One of the main limitations of this study is that it was conducted in vitro on bi-layered polyurethane blocks. Although this method has allowed to standardize the experimentation, it does not allow to perceive how the two systems would behave in vivo where there would be factors such as sex and age to influence the tightness of the miniscrews under load. It is our intention, for the future, to be able to study the mechanical behavior of the two systems examined also on patients to obtain data of even more support for clinical activity.

Further studies are therefore needed, including analyzing different anatomical sites or using different materials and configurations.

## 5. Conclusions

Taking into account the limitation that our study was developed with in vitro modeling and not in a clinical setting, it can be concluded that:a two-TAD system is on average stiffer than a one-TAD system under orthodontic loading;a one-TAD loses stability before reaching the YL of the TMA-archwire;the highest MLVs can be reached only with a two-TAD system; and the MLVs depends strictly on the archwire material in the two-TAD system.

## Figures and Tables

**Figure 1 jfb-11-00080-f001:**
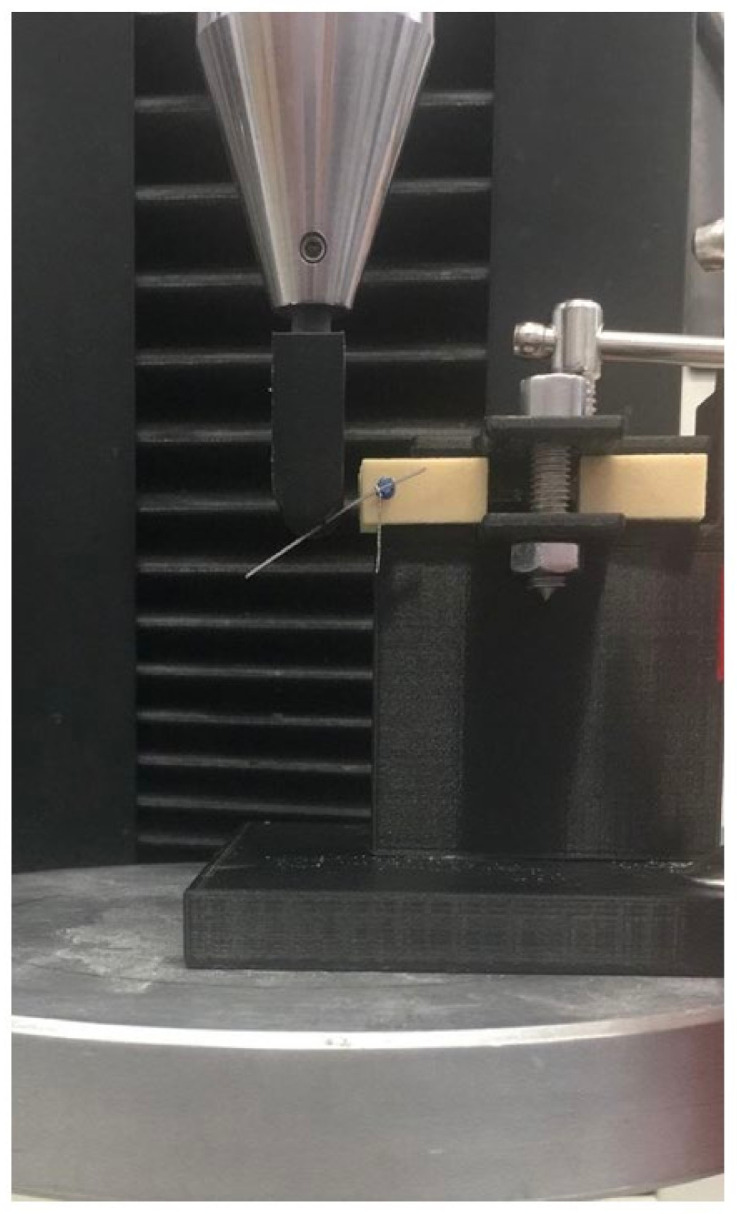
Mechanical test of one-miniscrew.

**Figure 2 jfb-11-00080-f002:**
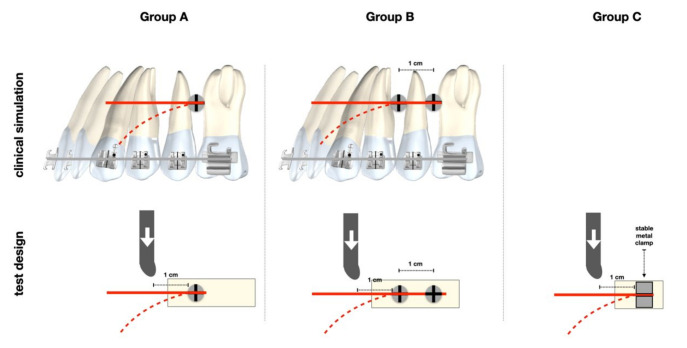
Test design and clinical simulation.

**Figure 3 jfb-11-00080-f003:**
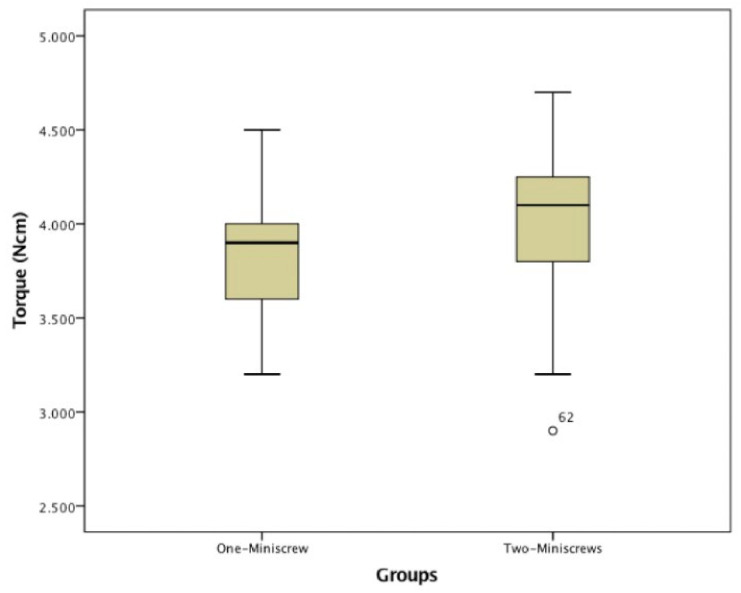
Boxplot distribution Insertion Torque Values.

**Figure 4 jfb-11-00080-f004:**
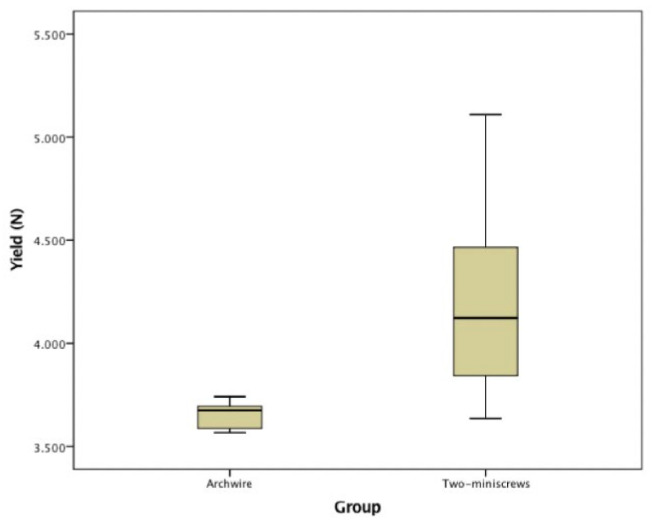
Boxplot Yield Points Group B and Group C.

**Figure 5 jfb-11-00080-f005:**
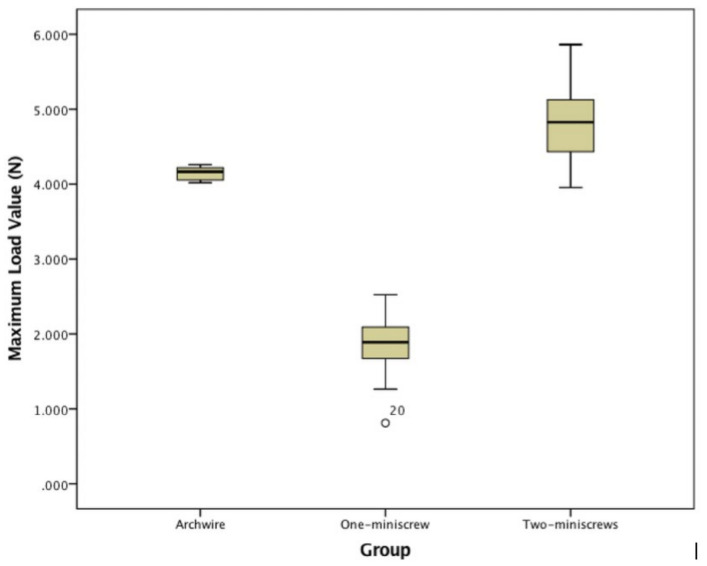
Boxplot Maximum Load Values of Group A, B, C.

**Figure 6 jfb-11-00080-f006:**
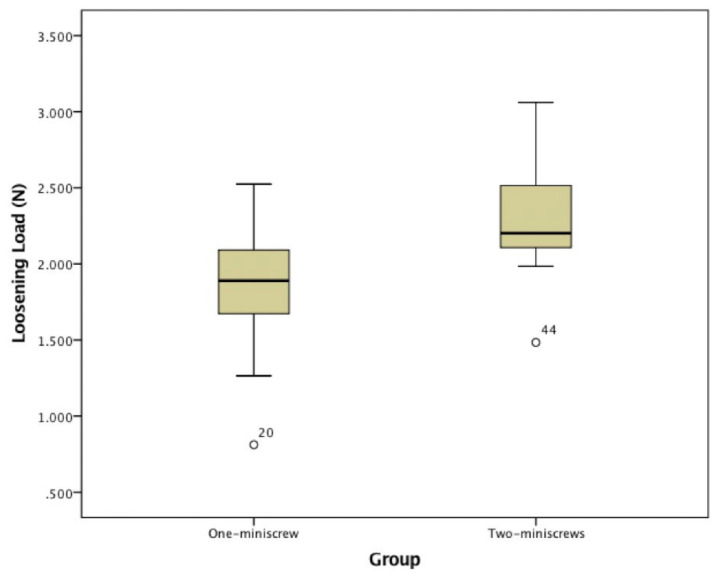
Boxplot Loosening Load of Group A and Group B.

**Figure 7 jfb-11-00080-f007:**
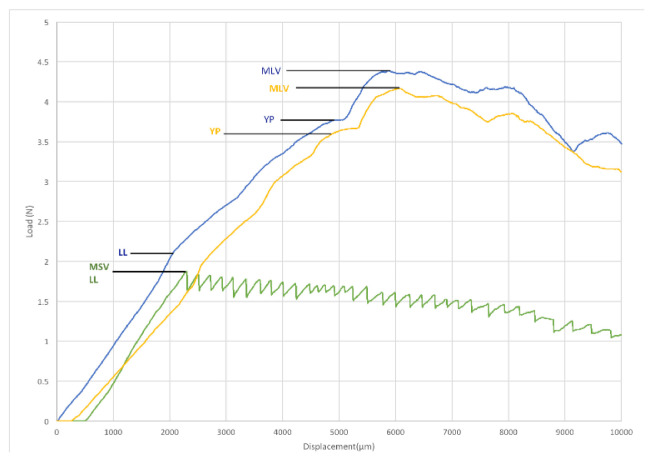
Load-Displacement Graph.

**Table 1 jfb-11-00080-t001:** Mann-Whitney Test for Insertion Torque Values.

	One-Miniscrew (23)		Two-Miniscrews (44)		
	Mean Value	SD	Mean Value	SD	*p*
Insertion Torque (Ncm)	3.870	0.388	4.014	0.399	0.089

**Table 2 jfb-11-00080-t002:** Main Value for Yield Load, Maximum Load Value, Loosening Load for Group A, B and C.

	One-Miniscrew (23 Singles)		Two-Miniscrews (22 Couples)		Archwire Only		
	Mean Value	SD	Mean Value	SD	Mean Value	SD	*p*
Yield (N)	NA	NA	4.189 N	±0.390 N	3.652 N	±0.064 N	0.000
Maximum Load Value (N)	1.871 N	±0.318 N	4.843 N	±0.515 N	4.150 N	±0.086 N	0.000
Loosening Load (N)	1.871 N	±0.318N	2.294 N	±0.333 N	NA	NA	0.000

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
