# Peer review of "Mechanical Evaluation of the Stability of One or Two Miniscrews under Loading on Synthetic Bone"

_jfb, 2020, doi:10.3390/jfb11040080_

Round 1

Reviewer 1 Report

Point 01

The manuscript needs a revision of English. Some examples:

“a two-miniscrews system” should be “a two-miniscrew system”

“The aim of the present study is” should be “The aim of the present study was”

“this is the first study that investigates” should be “this is the first study that investigated”

“The null hypothesis is” should be “The null hypothesis was”

“when sub-optimal condition are showed” should be “when sub-optimal conditions are showed”

“the loosening of a single miniscrew is may expected” should be “loosening of a single miniscrew is expected”

Point 02

In table 1 the mean ranks are presented. What one needs to present is not the ‘mean rank’, but the ‘mean value’ and standard deviation. And what is the point to present the ‘sum of ranks’, ‘Z’, ‘Mann-Whitney U’, and ‘Wilcoxon W’, if the authors are not using this information in the context of the analysis? The same is true for tables 2 and 3.

Point 03

There are several issues in the Discussion:

  • The first four paragraphs of the Discussion is not a discussion of the findings of the study. They consist of a short literature review, which should either be moved to the Introduction (if the subject was not already mentioned there) or removed from the text. There is also an overuse of references in these paragraphs;
  • The ninth paragraph of the Discussion is a pure repetition of the Results, without an actual discussion of the findings;
  • The tenth paragraph and the Figure 7 should be in the Results, not in the Discussion;
  • “The unscrewing problem can also be reduced using clockwise and anticlockwise TADs so that the orthodontic mechanics would not unscrew the TADs, although the armamentarium would be doubled.”: This was neither tested nor observed by the study;
  • Define “sub-optimal condition”;
  • No comparisons with the results of other studies (using one screw) were made.

All in all, the Discussion is very poor. This section seems more like a repetition of the Results than a discussion per se. It is amazing that the manuscript has 54 references but none of them were used to discuss the findings of the study. The authors should address this issue, and this will be of paramount importance for me to consider this manuscript for publication when I review the revised version of it.

Author Response

Point 01

The manuscript needs a revision of English. Some examples:

“a two-miniscrews system” should be “a two-miniscrew system”

 “The aim of the present study is” should be “The aim of the present study was”

“this is the first study that investigates” should be “this is the first study that investigated”

“The null hypothesis is” should be “The null hypothesis was”

“when sub-optimal condition are showed” should be “when sub-optimal conditions are showed”

“the loosening of a single miniscrew is may expected” should be “loosening of a single miniscrew is expected

A: Thank you for your suggestion. We have done it

Point 02

In table 1 the mean ranks are presented. What one needs to present is not the ‘mean rank’, but the ‘mean value’ and standard deviation. And what is the point to present the ‘sum of ranks’, ‘Z’, ‘Mann-Whitney U’, and ‘Wilcoxon W’, if the authors are not using this information in the context of the analysis? The same is true for tables 2 and 3.

A: Thank you for your suggestion. We have modified tables 1 and 2 and removed 3 and 4

Point 03

There are several issues in the Discussion:

  • The first four paragraphs of the Discussion is not a discussion of the findings of the study. They consist of a short literature review, which should either be moved to the Introduction (if the subject was not already mentioned there) or removed from the text. There is also an overuse of references in these paragraphs;

A: Thank you for your suggestion. We have deleted the indicated paragraph and removed many of the references in this section.

  • The ninth paragraph of the Discussion is a pure repetition of the Results, without an actual discussion of the findings;
  • The tenth paragraph and the Figure 7 should be in the Results, not in the Discussion

A: Thank you for your suggestion. We have deleted the ninth paragraph and removed some phrases in results section.

  • “The unscrewing problem can also be reduced using clockwise and anticlockwise TADs so that the orthodontic mechanics would not unscrew the TADs, although the armamentarium would be doubled.”: This was neither tested nor observed by the study;

A: Thank you for your suggestion. We added in the text that this sentence is our opinion and consideration that emerged observing the results of this study.

  • Define “sub-optimal condition”;

A: Thank you for your suggestion. We have replaced the term “sub-optimal“ with “detailed”.

  • No comparisons with the results of other studies (using one screw) were made.

A: Thank you for your suggestion. As already clarified in the text, from what emerges from the literature, this is the first research that compares systems with one and two mini-screws. Therefore it is difficult to compare with other studies.

All in all, the Discussion is very poor. This section seems more like a repetition of the Results than a discussion per se. It is amazing that the manuscript has 54 references but none of them were used to discuss the findings of the study. The authors should address this issue, and this will be of paramount importance for me to consider this manuscript for publication when I review the revised version of it.

A: Thank you for your suggestion. We have revised the whole discussion paragraph and ended several changes. Moreover we have removed 14 references.

Reviewer 2 Report

The manuscript performed by Pradal et al. evaluate the primary stability of a two-miniscrews system inserted into a synthetic bone. It provide some uesful informaiton for the readers.  Some minor suggestion are including as follows:

1 What are the materilas type of miniscrew used? 

2 Some preivous literature regarind bone screw are suggested for the author.

Mg bone implant: Features, developments and perspectives. Materials & Design, 2020, 108259

Author Response

1 What are the materilas type of miniscrew used? 

A: Thanks for the question: we have specified in the text that they are titanium

2 Some preivous literature regarind bone screw are suggested for the author.

Mg bone implant: Features, developments and perspectives. Materials & Design, 2020, 108259

A: Thank you for your suggestion. We have added the indicated reference.

Reviewer 3 Report

This manuscript compares mechanical properties among three experimental groups. The study idea is interesting and the experimental set-up is well-prepared. But some description and explanation should be clarified and revised.

  1. abstract, why do the values of yield load (YL) only presented in group B and C?
  2. materials and methods, "The settings for the motor were a uniform speed of 50 rpm and a torque limiter of 10 N∙cm. All insertions were recorded via USB (Table 1). "  However, in table 1,  why the insertion torque one-miniscrew group is 28.41 and two-miniscrew group is 36.92? It is poorly understood and inconsistent. Moreover, more descriptions of data should be addressed in the results. For example, the insertion torque of two-miniscrew group is calculated by the total values of 2 miniscrew or other methods.
  3. materials and methods, in the figure 4 and table 2, why is there lack of one screw group? The same problem is found in Fig.6.
  4. discussion, please cite the references"Evaluation of the implant stability and the marginal bone level changes during the first three months of dental implant healing process: A prospective clinical study. Su et al. J Mech Behav Biomed Mater. 2020 Oct;110:103899. 
  5. Please extend the more description in the result section.
  6. references, please delete the extra words. For example"Kuroda S, Sugawara Y, Deguchi T, Kyung H-M, Takano-Yamamoto T. Clinical use of miniscrew implants as orthodontic anchorage: Success rates and postoperative discomfort. Am J Orthod Dentofacial Orthop. gennaio 2007;131(1):9–15."

Author Response

abstract, why do the values of yield load (YL) only presented in group B and C?

A: Thanks for the question. As we have already reported in the text, being the YL “the point on a stress-strain or, as in this case, on the load-displacement curve, that indicates the end of the elastic region and the beginning of plastic region”, it cannot be calculated since the archwire maintained an elastic profile throughout the test in group A. This is evident in the graph in figure 7.

materials and methods, "The settings for the motor were a uniform speed of 50 rpm and a torque limiter of 10 N∙cm. All insertions were recorded via USB (Table 1). "  However, in table 1,  why the insertion torque one-miniscrew group is 28.41 and two-miniscrew group is 36.92? It is poorly understood and inconsistent.

A: Thanks for the question. The data reported in Tab 1 are the insertion torque recorded by the motor during the procedure. Although the device has been set to a definite torque value, factors such as the operator pressure, the insertion axis and the characteristics of the receiving site (although standard, as in this study, being bi-layered polyurethane blocks), can influence the insertion torque.

Moreover, more descriptions of data should be addressed in the results. For example, the insertion torque of two-miniscrew group is calculated by the total values of 2 miniscrew or other methods.

A: Thank you for your suggestion. We have expanded the results with a greater description of the data.

The insertion torque of two-miniscrew group reported the average of the values of each screws. We have added this description in the text.

materials and methods, in the figure 4 and table 2, why is there lack of one screw group? The same problem is found in Fig.6.

A: Thanks for the question. We have already explained previously that for group A (one-screw) it was not possible to calculate the Yeld, which is why it does not appear in figure 4. while the loosening load was not calculated in group C (archwire) because the value is referred to a property of screws not present in this group. this is why this data is not shown in figure 6.

discussion, please cite the references "Evaluation of the implant stability and the marginal bone level changes during the first three months of dental implant healing process: A prospective clinical study. Su et al. J Mech Behav Biomed Mater. 2020 Oct;110:103899.

A: Thank you for your suggestion. We have done it

Please extend the more description in the result section.

A: Thank you for your suggestion. We have done it

references, please delete the extra words. For example"Kuroda S, Sugawara Y, Deguchi T, Kyung H-M, Takano-Yamamoto T. Clinical use of miniscrew implants as orthodontic anchorage: Success rates and postoperative discomfort. Am J Orthod Dentofacial Orthop. gennaio 2007;131(1):9–15."

A: Thank you for your suggestion. We have done it

Round 2

Reviewer 1 Report

Point 01

The authors need to also present the standard deviation for the insertion torque values presented in Table 1.

Point 02

The authors need to explain in details how was the test setting for group C (archwire-only). There is a figure (Figure 2) well illustrating the test settings for groups A and B:

"The load was applied perpendicularly to the archwire at a distance of 10 mm from the center of the proximal TAD for Groups A and B (Figure 2). And from the proximal point of the clamp for Group C."

Which clamp? Where was this clamp? How was the archwire set in relation to this clamp? Is there a way to better illustrate/explain this?

Point 03

"this is the first research that compares systems with one and two mini-screws"

OK, but what about the results of other studies using only one screw?

Author Response

Point 01

The authors need to also present the standard deviation for the insertion torque values presented in Table 1.

A: Thank you for your suggestion. We have done it

Point 02

The authors need to explain in details how was the test setting for group C (archwire-only). There is a figure (Figure 2) well illustrating the test settings for groups A and B: "The load was applied perpendicularly to the archwire at a distance of 10 mm from the center of the proximal TAD for Groups A and B (Figure 2). And from the proximal point of the clamp for Group C." Which clamp? Where was this clamp? How was the archwire set in relation to this clamp? Is there a way to better illustrate/explain this?

A: Thank you for your suggestion. We have added more details in the text to clarify the test in group C and we have modified Figure 2. “For Group C, a custom-made clamp was designed and printed with an Ultimaker 3 Extended (Ultimaker, Utrecht, Netherlands) using polylactic acid and polyvinyl acid as copolymer. The resolution of the 3D printer was 0.4 mm. Subsequently, the clamp was fixed on a rigid support and a segment of 30 mm of rectangular 19” x 25” TMA archwire were tied within with stainless steel ligatures”

Point 03

"this is the first research that compares systems with one and two mini-screws" OK, but what about the results of other studies using only one screw?

A: Thank you for your request. We have already stated in the introduction that the primary stability of the single screw system is connected to several factors but we preferred to add a further sentence in the discussions that would relate with the results of our study: “Literature review has showed that the researches, relating to the loss of primary stability of mini-screws, no analyzes the unscrewing induced by the force applied to the wire but performed only pull out (5, 11), bend (23) or fracture tests (12,15). So they are not comparable with what has been analyzed and tested in our work because they had effort on different mechanics.Therefore, when we consider the primary stability of a mini-screw, we must relate it to different factors such: bone density (6, 10), cortical thickness (5, 21), the inclination of insertion (7, 11) and insertion torque (17). The influence of many of these factors can, hence, as emerges from our study, be minimized with a two-miniscrew system, where load and torsion forces are best issued”.

Reviewer 3 Report

Agree 

Author Response

Comments and Suggestions for Authors: Agree

A: thanks for your comments and suggestions. We have further improved the manuscript.